# Federated Learning with Nonvacuous Generalisation Bounds

## Abstract

We introduce a novel strategy to train randomised predictors in federated learning, where each node of the network aims at preserving its privacy by releasing a local predictor but keeping secret its training dataset with respect to the other nodes. We then build a global randomised predictor which inherits the properties of the local private predictors in the sense of a PAC-Bayesian generalisation bound. We consider the synchronous case where all nodes share the same training objective (derived from a generalisation bound), and the heterogenous and homogenous cases where each node may have its own personalised training objective. We show through a series of numerical experiments that our approach achieves a comparable predictive performance to that of the batch approach where all datasets are shared across nodes. Moreover the predictors are supported by numerically nonvacuous generalisation bounds while preserving privacy for each node. We explicitly compute the increment on predictive performance and generalisation bounds for our two federated settings, highlighting the price to pay to preserve privacy.

## 1 Introduction

In the federated learning (FL) paradigm, a group of *users* (or nodes) is learning in parallel, and typically aims at preserving their personal datasets while sharing a common predictor. While maintaining the privacy of their own data, users mutually share information through a central *server*. There has been a significant surge of interest in federated learning in the past decade (Konečný et al., 2016b), with clear applications in healthcare, transportation and retail, where it is typically of the utmost interest to avoid the leak of private information to other organisations or devices, for ethical or business motivations. The existing literature essentially categorises *horizontal* and *vertical* FL, depending on whether users' datasets share many features or individuals. These two streams have generated various contributions such as the design of efficient communication strategies (Konečný et al., 2016a;b; Suresh et al., 2017), the preservation of privacy through differentially-private distributed optimisation methods (Agarwal et al., 2018), the enforcement of fairness (as in many cases, post-training learning models may be biased or unfair and may discriminate against some protected groups – Hardt et al., 2016; Mohri et al., 2019). We refer to Zhang et al. (2021); Mammen (2021); Kairouz et al. (2021) for recent surveys on FL.

Consider a simple federated learning framework (Bonawitz et al., 2017; McMahan et al., 2017b). In each round, the server first provides the initial model to each user, then each user updates the initial model with its personal data. Finally, the server aggregates the collected local models into a single global model, which is used as next round's initialisation if needed. Hereafter we will refer to this learning problem as *FL-SOB* (Federated Learning with Synchronous OBjectives). This is especially relevant when all users share a common learning goal (*e.g.*, hospitals learning from different datasets to identify or predict a specific single pathology). Deep neural networks have been used to develop powerful federated algorithms (McMahan et al., 2017a). A more complex scenario consists in *personalised FL* (*PFL*, Tan et al., 2022) where users may have their own distinct learning goal but still want to share joint information as these goals share some level of similarity. This corresponds, for instance, to *transfer learning* (see *e.g.*, Zhuang et al., 2021) situations where one wants to extract some information of a learning problem (*e.g.*, detecting tigers in images) to perform better on another one, sharing some similarities (*e.g.*, detecting cats).

**Towards a unified framework.** The recent PAC-FL framework of Zhang et al. (2023b) proposes a unified framework to formalise FL, intricating the notion of generalisation ability (designed as utility) alongside privacy, and quantifying how much data are protected (*i.e.* impossible to retrieve) while transmitting partial information to the server. This framework builds from the work of Zhang et al. (2019) investigating the tradeoffs between privacy, utility and efficiency. The question of an optimal trade-off is crucial to deploy the FL framework in practice (Tsipras et al., 2019).

**On the place of generalisation in FL.** Using their PAC-FL framework, Zhang et al. (2023b) proposed generalisation bounds involving the dimension of the predictor space. The question of generalisation in FL is central: Mohri et al. (2019); Zhang et al. (2023a) established Rademacher-based generalisation bounds, Yagli et al. (2020) provided bounds based on mutual information to explain both generalisation ability and privacy leakage per user. Bayesian methods have also been considered in FL-SOB (Yurochkin et al., 2019; Chen & Chao, 2021; Zhang et al., 2022) as well as in PFL (Kotelevskii et al., 2022).

**PAC-Bayes learning in FL.** Beyond Bayesian methods, PAC-Bayes learning (see the seminal works of Shawe-Taylor & Williamson, 1997; McAllester, 1998; 2003; Maurer, 2004 – we refer to the surveys of Guedj, 2019; Alquier, 2021, and to the recent monograph of Hellström et al., 2023) has recently re-emerged as a powerful framework in batch learning to explain the generalisation ability of neural nets by providing non-vacuous generalisation bounds (Dziugaite & Roy, 2017; Letarte et al., 2019; Pérez-Ortiz et al., 2021; Biggs & Guedj, 2021; 2022). PAC-Bayes combines information-theoretic tools with the Bayesian paradigm of generating a data-dependent *posterior* distribution over a predictor space from a *prior* distribution (or reference measure), usually data-independent. The flexibility of the PAC-Bayes framework makes it useful to explain generalisation in many learning settings. In particular, theoretical results and practical algorithms have been derived for various learning problems such as reinforcement learning (Fard & Pineau, 2010), online learning (Li et al., 2018; Haddouche & Guedj, 2022), constrative learning (Nozawa et al., 2020), generative models (Chérief-Abdellatif et al., 2022), multi-armed bandits (Seldin et al., 2011; 2012; Sakhi et al., 2022), meta-learning (Amit & Meir, 2018; Farid & Majumdar, 2021; Rothfuss et al., 2021; 2022; Ding et al., 2021), majority votes (Zantedeschi et al., 2021; Biggs et al., 2022) to name but a few.

Recently, some works have used the PAC-Bayes framework in FL: Reisizadeh et al. (2020) and Achituve et al. (2021) have evaluated the post-training predictor shared by all users through a PAC-Bayes bound. Rather than exploiting existing bounds, new PAC-Bayes results, tailored for personalised FL, recently emerged with the aim to explain the efficiency of learning procedures (Scott et al., 2023; Sefidgaran et al., 2023), although the PAC-Bayes bound is not minimised by the algorithm. Finally, recent works showed that the Bayesian procedure ELBO, adapted to FL, is exactly the minimisation of a PAC-Bayes upper bound (Kim & Hospedales, 2023; Vedadi et al., 2023). Thus, they show that those methods are well incorporated in a theoretical framework explaining their good generalisation ability.

**Our contributions.** Beyond being a safety check for generalisation, PAC-Bayes theory provides state-of-the art learning algorithms with tight generalisation guarantees in the batch setting. We adapt those algorithms to the FL-SOB and PFL settings. We propose GENFL (standing for Generalisation-driven Federated Learning), an algorithm in which users optimise local PAC-Bayes objectives (bounds from Dziugaite & Roy, 2017; Pérez-Ortiz et al., 2021). We show a global generalisation bound for all users in FL-SOB, and local ones in PFL. Finally, we show in numerical experiments that our procedure is competitive with the state-of-the-art and we bring nonvacuous generalisation guarantees to practitioners of federated learning.

**Outline.** We describe our notation in Section 2 and introduce in Section 3 a novel algorithm called GENFL, alongside two instantiations to FL-SOB and PFL. We present numerical experiments to support our methods. in Section 4. Our algorithms and the code used to generate figures in this paper is available at https://anonymous.4open.science/r/GenFL-0147/README.md. The paper closes with a discussion in Section 5. In Appendix A, we comment on strategies to compute PAC-Bayesian bounds, Appendix B contains a comprehensive description of our procedure in the PFL setting, and Appendix C provides additional experiments.

## 2 Background

**Federated learning.** We consider a predictor set $\mathcal{H}$, a data space $\mathcal{Z}$ and denote the space of distributions over $\mathcal{H}$, $\mathcal{M}(\mathcal{H})$. We let $\ell \colon \mathcal{H} \times \mathcal{Z} \to [0,1]$ denote a loss function. In FL, we consider an ensemble of $K \in \mathbb{N}^*$ users, and for each user $1 \le i \le K$, we denote by $\mathcal{S}_i = (\mathbf{z}_{i,j})_{j=1\cdots m_i}$ its associated dataset of size $m_i$. We define $\mathcal{S}$, of size $m = \sum_{i=1}^{K} m_i$, to be the union of all $\mathcal{S}_i$. We assume that each $\mathcal{S}_i$ is *i.i.d.* with associated distribution $\mathcal{D}_i$. Each user $1 \le i \le K$ aims to jointly learn a predictor $h \in \mathcal{H}$ while keeping private their training dataset $\mathcal{S}_i$.

**Learning theory.** In PAC-Bayes learning, instead of directly crafting a predictor $h \in \mathcal{H}$, we design a data-driven posterior distribution $Q \in \mathcal{M}(\mathcal{H})$ with respect to a prior distribution $P$. To assess the generalisation ability of a predictor $h \in \mathcal{H}$, we define for each user $k$ the *risk* to be $R_{\mathcal{D}_i} := \mathbb{E}_{\mathbf{z} \sim \mu}[\ell(h, \mathbf{z})]$ and its empirical counterpart $\hat{R}_{\mathcal{S}_i} := \frac{1}{m} \sum_{j=1}^{m_i} \ell(h, \mathbf{z}_{i,j})$. As PAC-Bayes focuses on elements of $\mathcal{M}(\mathcal{H})$, we also define the expected risk and empirical risks for $Q \in \mathcal{M}(\mathcal{H})$ as $R_{\mathcal{D}_i}(Q) := \mathbb{E}_{h \sim Q}[R_{\mathcal{D}_i}(h)]$ and $\hat{R}_{\mathcal{S}_i}(Q) := \mathbb{E}_{h \sim Q}[\hat{R}_{\mathcal{S}_i}(h)]$.

**Background on PAC-Bayes learning.** In a batch setting, we only consider the dataset $\mathcal{S}$ (this can be seen as the case where there is only one user) and we assume that all data are *i.i.d.* with distribution $\mathcal{D}$. For two probability measures $P, Q$ we define the *Kullback-Leibler divergence* to be $\mathrm{KL}(Q, P) = \mathbb{E}_{h \sim P}\left[\frac{dQ}{dP}(h)\right]$ where $\frac{dQ}{dP}$ is the Radon-Nikodym derivative. We also denote by kl the KL divergence between two Bernoulli distributions.

**Generalisation bounds.** We recall the following bound, due to McAllester (2003); Maurer (2004), which holds for bounded losses.

**Theorem 1** (McAllester's bound)**.** *For any data-free prior distribution* $P \in \mathcal{M}(\mathcal{H})$*, any* $\delta \in [0,1]$*, with probability at least* $1 - \delta$*, for any posterior distribution* $Q \in \mathcal{M}(\mathcal{H})$*,*

$$\mathrm{kl}\left(R_{\mathcal{D}}(Q), R_{\mathcal{S}}(Q)\right) \le \frac{\mathrm{KL}(Q\|P) + \ln\frac{2\sqrt{m}}{\delta}}{m}, \tag{1}$$

*which leads to the following upper bound on the risk*

$$R_{\mathcal{D}}(Q) \le \mathrm{kl}^{-1}\left(\hat{R}_{\mathcal{S}}(Q) \,\middle\|\, \frac{\mathrm{KL}(Q\|P) + \ln\frac{2\sqrt{m}}{\delta}}{m}\right), \tag{2}$$

*where* $\mathrm{kl}^{-1}(x, b) = \sup\{y \in [x, 1] \mid kl(x, y) \le b\}$*.*

Note that by the definition of $kl^{-1}$, (2) is the tightest upper bound on $R_{\mathcal{D}}(Q)$ that we can obtain starting from (1). While $\mathrm{kl}^{-1}$ has no closed form, it is possible to approximate it efficiently via root-finding techniques (see, *e.g.*, Dziugaite & Roy, 2017, Appendix A). However, this function is hard to evaluate, and even harder to optimise. We need to rely on looser relaxations of (1) to design tractable optimisation procedures.

**Relaxations of McAllester's bound.** The most classical relaxation of (1) relies on Pinsker's inequality $\mathrm{kl}(q\|p) \ge \frac{(p-q)^2}{2}$ and leads to the following high-probability bound, valid under the same assumptions as Theorem 1:

$$R_{\mathcal{D}}(Q) \le \hat{R}_{\mathcal{S}}(Q) + \sqrt{\frac{\mathrm{KL}(Q\|P) + \ln\frac{2\sqrt{m}}{\delta}}{2m}}. \tag{3}$$

While (3) is well known and already appears in McAllester (2003), novel relaxations, exploiting refined Pinsker's inequality (see, *e.g.*, Boucheron et al., 2013, Lemma 8.4) have been exploited to obtain PAC-Bayes Bernstein bounds (Tolstikhin & Seldin, 2013; Mhammedi et al., 2019). Building on this inequality, Rivasplata et al. (2019); Pérez-Ortiz et al. (2021) proposed a *PAC-Bayes quadratic bound* recalled below, valid under

the assumptions of Theorem 1

$$R_{\mathcal{D}}(Q) \leq \left( \sqrt{\hat{R}_{\mathcal{S}}(Q) + \frac{KL(Q\|P) + \log\left(\frac{2\sqrt{m}}{\delta}\right)}{2m}} + \sqrt{\frac{KL(Q\|P) + \log\left(\frac{2\sqrt{m}}{\delta}\right)}{2m}} \right)^2. \tag{4}$$

Note that (3) and (4) are easier to optimise than (2), making them more relevant for practical learning algorithms.

**Generalisation-driven learning algorithms.** Most of the PAC-Bayesian bounds in the literature are fully empirical. This paves the way to use the bound as a training objectives and leads to generalisation-driven learning algorithms. A classical PAC-Bayesian algorithm is derived from Catoni's bound (see, *e.g.*, Catoni, 2007, Alquier et al., 2016, Theorem 4.1):

$$\underset{Q \in \mathcal{M}(\mathcal{H})}{\operatorname{argmin}} R_{\mathcal{S}}(Q) + \frac{KL(Q, P)}{\lambda}. \tag{5}$$

In (5) an *inverse temperature* $\lambda > 0$ appears and acts as a learning rate in gradient descent. Similarly, it is possible to derive batch learning algorithms from (3) (4) (Dziugaite & Roy, 2017; Pérez-Ortiz et al., 2021). Then, we have access to a theoretical upper bound which requires to approximate expectations over Q. We next discuss how to mitigate this.

**Computing generalisation guarantees.** In practice, the tightest McAllester's bound is computed, *i.e.*, (1). First, note that the KL divergence is easy to compute in the Gaussian case as it has a closed form (see, *e.g.*, Duchi, 2007, Section 9). Then, it remains to estimate the expected empirical risk over Q, which is costly in practice as it involves Monte Carlo approximations. To alleviate this issue, we leverage the trick from Dziugaite & Roy (2017, Section 3.3), which exploit a high probability upper bound of $\hat{R}_{\mathcal{S}}(Q)$ with confidence level $\delta'$

$$\hat{R}_{\mathcal{S}}(Q) \leq kl^{-1}\left( \hat{R}_{\mathcal{S}}(\hat{Q}_n) \left\| \frac{1}{n} \ln\left(\frac{2}{\delta'}\right) \right. \right),$$

where $\hat{R}_{\mathcal{S}}(\hat{Q}_n) = \frac{1}{n} \sum_{i=1}^{n} \ell(W_i, z_i), \forall i, W_i \sim Q$. Then incorporating this upper bound in (1) gives the final bound we use, with probability at least $1 - \delta - \delta'$:

$$R_{\mathcal{D}}(Q) \leq kl^{-1}\left( \hat{R}_{\mathcal{S}}^{n}(Q) \left\| \frac{KL(Q\|P) + \ln\frac{2\sqrt{m}}{\delta}}{m} \right. \right), \tag{6}$$

where $\hat{R}_{\mathcal{S}}^{n}(Q) = kl^{-1}\left( \hat{R}_{\mathcal{S}}(\hat{Q}_n) \left\| \frac{1}{n} \ln\left(\frac{2}{\delta'}\right) \right. \right)$.

To compute $kl^{-1}(p, c)$ for any $p, c$, we leverage Dziugaite & Roy (2017, Appendix A) described in appendix A.

## 3 Generalisation-driven Federated Learning

In this section we introduce our algorithm GENFL.

**From batch to federated PAC-Bayes algorithms.** When training stochastic neural nets (SNNs) with PAC-Bayes objectives, it is common to assume that each weight follows a Gaussian distribution. For conciseness, we identify a SNN to the Gaussian distribution of all its weights $\mathcal{N}(\mu, Diag(\sigma_i))$. The works of Dziugaite & Roy (2017); Rivasplata et al. (2019); Biggs & Guedj (2021); Pérez-Ortiz et al. (2021); Perez-Ortiz et al. (2021); Biggs & Guedj (2022) proposed successful PAC-Bayesian training algorithms for SNNs which ensure generalisation guarantees. All these methods operate in a batch setting, *i.e.*, the optimiser has access to all data simultaneously. Thus, building on the work of Rivasplata et al. (2019); Pérez-Ortiz et al.

(2021), we propose Alg. 1, a new learning algorithm called GENFL (Generalisation-driven Federated Learning), casting PAC-Bayes into FL. We stress that GENFL benefits from nonvacuous generalisation guarantees (Section 4).

---

**Algorithm 1** GENFL. users are indexed by $k$; $B$ is the local minibatch size, $E$ is the number of local epochs, $\eta$ is the learning rate, $f$ the PAC-Bayes objective. The prior is $\mathcal{N}(\mu_{\text{prior}}\sigma_{\text{prior}})$ with parameter $\delta$.

1: **Server executes:**
2: $m \leftarrow \sum_{k=1}^{K} m_k$             ▷ Total dataset size
3: $w_1 \leftarrow (\mu_{\text{prior}}, \sigma_{\text{prior}})$
4: **for** each round $t$ **do**
5:     $S_t \leftarrow$ random set of $\max(C \cdot K, 1)$ users
6:     **for** each user $k \in S_t$ **in parallel do**
7:        $w_{t+1}^k \leftarrow \text{userUpdate}(k, w_t, m)$
8:     **end for**
9:     $w_{t+1} \leftarrow \sum_{k=1}^{K} \frac{m_k}{m} w_{t+1}^k$
10: **end for**
11:
12: **userUpdate(k, w, m):**
13: $\mathcal{B} \leftarrow$ (split $\mathcal{S}_k$ into batches of size B)
14: **for** each local epoch $e = 1, 2, \cdots, E$ **do**
15:     **for** each local minibatch $b \in B$ **do**
16:        $w_s^k \leftarrow \mu^k + \sigma^k \odot \mathcal{N}(0, 1)$          ▷ Reparam. trick
17:        $w^k \leftarrow w^k - \eta \nabla_w f_{m, \delta, \mu_{\text{prior}}, \sigma_{\text{prior}}}(w_s^k; b)$
18:     **end for**
19: **end for**
20: **return** $w^k$
**Ensure:** Global model distribution $w_T$ mapped to $\mathcal{N}(\mu_T, \sigma_T)$

---

**A generalisation-driven FL algorithm.** GENFL combines a federated learning protocol (*i.e.*, FedSGD, FedAvg, McMahan et al. (2017a)) with a PAC-Bayes objective $f$. Its starts from the vector $w_1 = (\mu_{prior}, \sigma_{prior})$ corresponding to the initial distribution $\text{P} = \mathcal{N}(\mu_{prior}, \sigma_{prior})$ and outputs after $T$ rounds $w_T = (\mu_T, \sigma_T)$ corresponding to the posterior $Q = \mathcal{N}(\mu_T, \sigma_T)$. Hence the learning procedure is divided in rounds, where subsets of users are sampled. Sampled users are requested to perform local updates following a PAC-Bayes training procedure designed to ensure a good generalisation of the posterior distribution. Because users train SNNs, they sample weights from the posterior. In order to learn the variance parameter of the posterior (Gaussian), we use the well-known reparameterisation trick (Kingma & Welling, 2014): instead of directly sampling from the distribution, we sample from a standard Gaussian distribution and then apply a transformation to obtain the sampled weights: $W = \mu + \sigma V$ where $V \sim \mathcal{N}(0, \text{Id})$, this allows to compute with respect to $\sigma$. When the round ends, the global model is computed from the local updates, thanks to the aggregation function from the FL protocol (weighted mean, median).

Next we show that with a PAC-Bayes objective $f$, we adapt GENFL to FL-SOB, where $\mathcal{S}$ is fully *i.i.d.*, *i.e.*, all user datasets have the same distribution, and to PFL where each dataset $\mathcal{S}_i$ is *i.i.d.* but any two dataset can have distinct distributions.

## 3.1 GenFL for FL-SOB

**PAC-Bayesian objectives.** We assume that for any $i$, $\mathcal{D}_i = \mathcal{D}$. Thus, $\mathcal{S}$ is a *i.i.d.* dataset of $m$ points. We then consider the true and empirical risks on all $\mathcal{S}$ $R_{\mathcal{D}}, R_{\mathcal{S}}$. In a batch setting, it would be natural to optimise the bounds (3), (4). However, the user $i$ only has access to its personal dataset $\mathcal{S}_i$ (of size $m_i$) to optimise its model *while knowing other datasets are involved.* We then derive accordingly from (3), (4) two PAC-Bayesian learning algorithms, valid for any user, namely $f_1, f_2$. Note that $f_1$ is adapted from the

$f_{classic}$ objective and $f_2$ is adapted from $f_{quad}$ of Pérez-Ortiz et al. (2021):

$$f_1(\mathcal{S}_i) = \hat{\mathrm{R}}_{\mathcal{S}_i}(Q) + \sqrt{\frac{\mathrm{KL}(Q\|P) + \ln\frac{2\sqrt{m}}{\delta}}{2m}}. \tag{7}$$

$$f_2(\mathcal{S}_i) = \left(\sqrt{\hat{\mathrm{R}}_{\mathcal{S}_i}(Q) + \frac{\mathrm{KL}(Q\|P) + \log\left(\frac{2\sqrt{m}}{\delta}\right)}{2m}} + \sqrt{\frac{\mathrm{KL}(Q\|P) + \log\left(\frac{2\sqrt{m}}{\delta}\right)}{2m}}\right)^2. \tag{8}$$

Note that (7), (8) can be seen as proxys of (3), (4). Indeed, every user has access to the total number of data points $m$ (as long as it is transmitted to the server), so the regularisation term (containing the KL divergence) is fully available, contrary to the empirical risk $\hat{\mathrm{R}}_{\mathcal{S}}$ which is then replaced by $\hat{\mathrm{R}}_{\mathcal{S}_i}$. Note that in this case, the KL divergence is divided by $m$ instead of $m_i$ (which would be natural if we were optimising (3) (4) for $\mathcal{S}_i$ instead of $\mathcal{S}$). This suggests that each user has to give more weight, during the optimisation phase, to its data than to the regularisation. The reason behind this is that the server, by aggregating predictors, performs a regularisation step on a global level, hence the need to prioritise data on a local one.

**A global generalisation guarantee.** A major interest of the *i.i.d.* assumption on $\mathcal{S}$ is that, as long as users all exploit the same posterior distribution $Q$, and they transmit their empirical scores $\hat{\mathrm{R}}_{\mathcal{S}_i}(Q)$, every user is able to compute the global generalisation guarantee of (2). This allows to maintain the bound of the batch setting (involving the total number of data points $m$), despite being in FL. This is empirically shown in Section 4. We present in Algorithm 2 FEDBOUND, the algorithm we use to compute the global bound (6), valid for all users simultaneously.

---

**Algorithm 2** FEDBOUND. The K users are indexed by k; $f$ PAC-Bayes objective, prior $\mathcal{N}(\mu_{\mathrm{prior}}, \sigma_{\mathrm{prior}})$; posterior $\mathcal{N}(\mu_T, \sigma_T)$, $\delta$, $\delta'$ parameters, $n$ number of Monte Carlo sampling

1: **Server executes:**
2: $m \leftarrow \sum_{k=1}^{K} m_k$          ▷ Total dataset size
3: $\mathrm{P} = \mathcal{N}(\mu_{\mathrm{prior}}, \sigma_{\mathrm{prior}})$          ▷ Prior (learned or random)
4: $\mathrm{Q} = \mathcal{N}(\mu_T, \sigma_T)$          ▷ Posterior (learned)
5: **for** each user $k \in K$ **in parallel do**
6:     $error^k \leftarrow \mathrm{userMCSampling}(k, w_t, m)$
7: **end for**
8: $error \leftarrow \sum_{k=1}^{K} \frac{m_k}{m} error_k$
9: $KL\_inv \leftarrow \mathrm{kl}^{-1}\left(error \mid \frac{1}{n}\ln(\frac{2}{\delta'})\right)$
10: Up-bound $\leftarrow \mathrm{kl}^{-1}\left(KL\_inv \mid \frac{\mathrm{KL}(Q\|P)+\ln\frac{2\sqrt{m}}{\delta}}{m}\right)$
11:
12: **userMCSampling(k, w, m):**
13: **for** each MC sampling $i = 1, 2, \cdots, n$ **do**
14:     $W_i^k \sim \mathrm{Q}$          ▷ Sample weights from the posterior
15:     $error_i^k \leftarrow \hat{\mathrm{R}}_{\mathcal{S}_k}(W_i^k)$          ▷ local empirical risk
16: **end for**
17: $error^k \leftarrow = \frac{1}{n}\sum_{i=1}^{n} error_i^k$
18: **return** $error^k$
**Ensure:** Up-bound holding with probability $1 - \delta - \delta'$

---

## 3.2 GenFL for personalised federated learning

**A general training for the prior distribution.** In PFL, the learning objective of each user may differ, while sharing some similarities that can be learned and transferred from one user to another. This framework requires adjustments of our learning objectives. Indeed, contrary to Section 3.1, there is no clear global generalisation guarantee, so each user has then to optimise its own personal learning objective from a

commonly shared prior. Using either a random prior or a learnt one on a fraction of users data, we run GENFL similarly to Section 3.1 with our PAC-Bayesian objective of interest $f$. The output distribution of GENFL is then considered as a common prior for all users which then needs to be personalised.

**A personalisation step.** Once a common prior distribution has been obtained from the federated training, it is necessary for each user to personalise it to its own problem. To do so, we apply PAC-Bayesian objectives similar to those in Section 3.1, namely $f_1$ (7) and $f_2$ (8), where the batch size is modified from $m$ to $m_i$ for the $i$-th user. This reflect that each user now optimises its local goal instead of the global one. Each user ends up with its own personalised posterior distribution. The way personalised bounds are implemented is similar to Algorithm 2, but without the aggregation step. We refer to Appendix B for additional details.

## 4 Experiments

In this section we provide practical instantiations of GENFL. We first consider in Section 4.1 the case of classification on MNIST in a federated setting using basic neural architectures. We then extend our experimental framework in Section 4.2 to the more challenging case of classification on CIFAR-10 with more sophisticated neural networks.

### 4.1 Classification on MNIST

#### 4.1.1 Experimental framework

Our experimental framework is inspired from Pérez-Ortiz et al. (2021) combined with the FedAvg protocol McMahan et al. (2017a). We use the following libraries: Pytorch (Paszke et al., 2019) for deep learning, Flower (Beutel et al., 2020) for federated learning, Slurm (Yoo et al., 2003) for cluster experiments, and Hydra (Yadan, 2019) for overall experiment management. The cluster nodes we use have 48 SKYLAKE 3GHz CPUS. We do not use GPUs.

**Prior distribution over weights.** We propose two types of priors: data-free (random) prior chosen randomly around $\mathcal{N}(0, \mathrm{Id})$ (as in Dziugaite & Roy, 2017) and a data-dependent (learnt) prior. The latter is powerful to attenuate the KL divergence term, leading to sharper generalisation bounds and better accuracy. Such a data-driven prior implies to use a fraction of the dataset from the training data to optimise the prior. The bound computation is then realised with a reduced dataset size (divided by 2 in practice). However, the prior has gained efficiency (lower empirical risk) and the PAC-Bayes optimisation starts from a relevant point.

We use Gaussian distribution for both prior and posterior over the weights of a neural network. When data-free, the prior is $P = \bigotimes_{l \in \text{layers}} \mathcal{N}(\text{truncated}(\mu_{\text{rand}}^l), \text{Diag}(\sigma_{\text{prior}}))$ with $\mu_{\text{rand}}^l \sim \mathcal{N}(0, \frac{1}{\sqrt{n_{in}^l}})$, and $\sigma_{\text{prior}} \in \mathbb{R}^{+*}$. The truncature is done at $\pm \frac{2}{\sqrt{n_{in}^l}}$ where $n_{in}^l$ is the dimension of the inputs of the layer $l$. In the case of data-dependent prior, we have $P = \bigotimes_{l \in \text{layers}} \mathcal{N}(\mu_{\text{learnt}}^l, \text{Diag}(\sigma_{\text{prior}}))$, where $\mu_{\text{learnt}}$. It is obtained via ERM on the prior set on half of the training set, the other half being used for bound computation and posterior optimisation.

**Dataset partition.** To build a *i.i.d.* FL setup, we consider the case where each user has exactly the same number of samples per class. We partition MNIST as follows: we fix the number of users to 100. Then, each user receives a dataset size of 540, each class having 54 images. In the case of the learnt prior, we split the training set of each user in half, the first one being used to train the prior, the second exploited by our learning algorithms.

In the case of non-*i.i.d.* FL setup, we follow McMahan et al. (2017a). First we sort MNIST by label, then we partition the dataset into chunks of 300 contiguous samples each (thus containing at most 2 labels, because it is sorted). Again we split each user dataset in several parts. When the prior is random, we save 10% of the dataset to create a validation set. Remaining data is exploited for optimisation. When considering learnt priors, we save again 10% of the dataset as a validation set, we exploit 40% of the dataset to train the prior (respecting the proportion of each class), the remaining 50% being used by our learning algorithms.

**Bound parameters.** We used $\delta = 0.05$, $\delta' = 0.01$, $n = 150000$ Monte Carlo samples.

**Optimisation hyperparameters.** The prior distribution scale $\sigma_{\mathrm{prior}}$ is set to $2, 5 \times 10^{-2}$, the learning rate is $5 \times 10^{-3}$ for 100 users. In order to compare with the batch learning setting, we compute our algorithms with 1 user. In this case, we use a learning rate of $5 \times 10^{-4}$ to reach better performances. The momentum is 0.95 for posterior optimisation and 0.99 in prior optimisation. During prior optimisation, we used a dropout rate of 0.2 to avoid overfitting. As theoretical results of Section 2 require a loss function in $[0,1]$, we use the bounded cross-entropy as in Pérez-Ortiz et al. (2021), *i.e.*, $\ell(x,y) = \frac{1}{\ln(p_{\min})} \cdot \ln(\tilde{\sigma}(x)_y)$ with $\tilde{\sigma}(x)_y = \max(\mathrm{sigmoid}(x)_y, p_{\min})$. We took $p_{\min} = 10^{-4}$.

**KL penalty.** For stability reasons, we penalise the KL term during posterior optimisation (similarly to Pérez-Ortiz et al., 2021), thus we give more impact to the empirical risk during optimisation. Such a penalty helps performance and stability during training when random priors are involved. In this case, we use a penalty of 0.1.

**Federated Learning hyperparameters.** Starting from a random prior, we perform our algorithm during 200 rounds to make the SNNs converge. When learnt priors are considered, they are trained with a run of 100 rounds with 5 local epochs (convergence around 50 epochs). We then perform 10 additional rounds with 5 local epochs to train the posterior. In both cases, we select 10% of the users each round to participate in the training. As the dataset size of each user is small, we use a batch size of 25 (compared to 250 in the work of Pérez-Ortiz et al. (2021)).

**Neural Architecture.** We consider a stochastic 2 hidden layer MLP with 600 units each, resulting in 1,198,210 number of parameters for the prior (with fixed covariance matrix) and doubled for the SNN (as we consider diagonal covariance matrices).

**Positive variance prior.** To have a constrained positive standard deviation $\sigma$ when sampling weights, we use the following transformation: $\sigma = \ln(1 + \exp(\rho))$. It makes $\sigma$ always positive, and $\rho$ can be any real number that is optimised during training procedure.

### 4.1.2 Results

Note that in a classification problem, the generalisation error translates a positive influence of the learning phase as long as it is smaller than 1 (which is what we refer to with the term nonvacuous). Indeed, a bound below this threshold shows that the posterior will not fail at each try. However, we focus on posteriors with generalisation bounds or test error smaller than 50% The reason is that, for a classification task, this threshold is the generalisation error of a randomised predictor with associated distribution Bernoulli(0.5). Thus, having results below this threshold provably show we generalise better than a naive strategy.

**FL-SOB setting.**

Table 1 gathers our results for GENFL applied with $f_1, f_2$ alongside FEDBOUND. We compared our results with 100 clients with the output of our algorithms for 1 client, corresponding to the batch learning case. Our FL algorithms benefit from nonvacuous theoretical guarantees and test errors. In the case of data-dependent priors, test errors of GENFL nearly reach for both $f_1$ and $f_2$, the precision of their batch counterpart (3% in FL and 2% in batch). In the case of random prior, the KL penalty has a strong positive influence on the test errors. The generalisation bounds of our algorithms are uniformly deteriorated compared to the batch setting, *while being of the same magnitude*. This is important to notice as this is the price to pay to adapt batch bounds to a federated setting. Indeed, as each user only optimised a proxy of the common generalisation bound, it is legitimate to retrieve in our results a short discrepancy comparing to the batch case.

While $f_2$ is consistently achieving better generalisation upper bounds in Pérez-Ortiz et al. (2021), it is outperformed by $f_1$ in the FL setting. However, notice that $f_2$ provides uniformly better test errors than $f_1$, similarly to Pérez-Ortiz et al. (2021). Note that better results are achieved if one considers the KL penalty trick with data-free prior and no KL penalty trick with data-dependent prior. We interpret this fact as follows: given that the data-dependent prior is already performing well on training data, allowing the posterior optimisation to be unconstrained is not an issue as we found an area close from a local minimiser.

Table 1: Results for the FL-SOB scenario. $\ell^{0-1}$ corresponds to the 0-1 loss. The test error column is made on the test set of MNIST. The Bound column corresponds to the generalisation bounds, computed with Alg. 2. The $KL/m$ column corresponds to the KL divergence term in the bound divided by $m = 60000$ in data-free prior or $m = 30000$ data-dependent prior.

| Setup | | Bound | Test Err. | KL div |
|---|---|---|---|---|
| Prior | Obj. | $\ell^{0-1}$ | $\ell^{0-1}$ | $KL/m$ |
| Pérez-Ortiz et al. (2021): Random | $f_1$ | 0.330 | 0,141 | 0,081 |
| (1 client) | $f_2$ | 0.316 | 0,092 | 0,138 |
| Pérez-Ortiz et al. (2021): Learnt | $f_1$ | 0.028 | 0,023 | <0,001 |
| (1 client) | $f_2$ | 0.028 | 0,020 | 0,001 |
| 100 users - GenFL - KL Penalty=0.1 | | | | |
| Random | $f_1$ | 0.333 | 0,123 | 0,107 |
| (us) | $f_2$ | 0.342 | 0,090 | 0,163 |
| Learnt | $f_1$ | 0,061 | 0,030 | <0,001 |
| (us) | $f_2$ | 0,088 | 0,029 | 0,002 |
| 100 users - GenFL - No KL Penalty | | | | |
| Random | $f_1$ | 0,415 | 0,256 | 0,039 |
| (us) | $f_2$ | 0,408 | 0,251 | 0,041 |
| Learnt | $f_1$ | 0.039 | 0,030 | <0,001 |
| (us) | $f_2$ | 0.040 | 0,030 | <0,001 |

Table 2: Results for the PFL scenario. $\ell^{0-1}$ corresponds to the 0-1 loss. The test error column is made on the test set of each user (10% of local dataset).The Gen. Bound column gathers generalisation bounds. Each user bound is computed locally with $m_i = 300$ for learnt prior, while $m_i = 540$ for random prior.

| Setup | | Gen. Bound $\ell^{0-1}$ | | |
|---|---|---|---|---|
| Prior | Obj. | min | mean | max |
| Random | $f_1$ | 0,063 | 0,680 | 0,847 |
| (us) | $f_2$ | 0,075 | 0,713 | 0,893 |
| Learnt | $f_1$ | 0,054 | 0,112 | 0,222 |
| (us) | $f_2$ | 0,052 | 0,111 | 0,220 |
| | | Test Error $\ell^{0-1}$ | | |
| Random | $f_1$ | 0 | 0,552 | 0,767 |
| (us) | $f_2$ | 0 | 0,588 | 0,833 |
| Learnt | $f_1$ | 0 | 0,050 | 0,183 |
| (us) | $f_2$ | 0 | 0,044 | 0,150 |

However, as the random prior is not necessarily efficient, we need to move far from it to reach good empirical performances. However, moving freely from the prior distribution leads to a large KL divergence, hence the need to constrain the posterior optimisation to obtain both better bounds and test errors. A take-home message is that adapting PAC-Bayes algorithms to FL is effective: it gives nonvacuous results close to the batch setting.

**PFL setting.**

Table 2 provides an overview of our results in the non-*i.i.d.* case. It gathers, for both generalisation bounds and test error, the minimum, mean and maximum performance of all 100 users. The averaged performances are deteriorated compared to Table 1 for all settings as we consider a harder problem. It is worth noticing that our bounds are nonvacuous and that our algorithms with learnt priors benefit from bounds and test errors lower than 50%. Indeed, if we do not learn the prior, we see from the distribution of errors in Figure 1 that most users have a deteriorated bound and test errors. For learnt priors, the error distribution shows that all users enjoy a meaningful bound as well as sound performance. An interesting point is that the

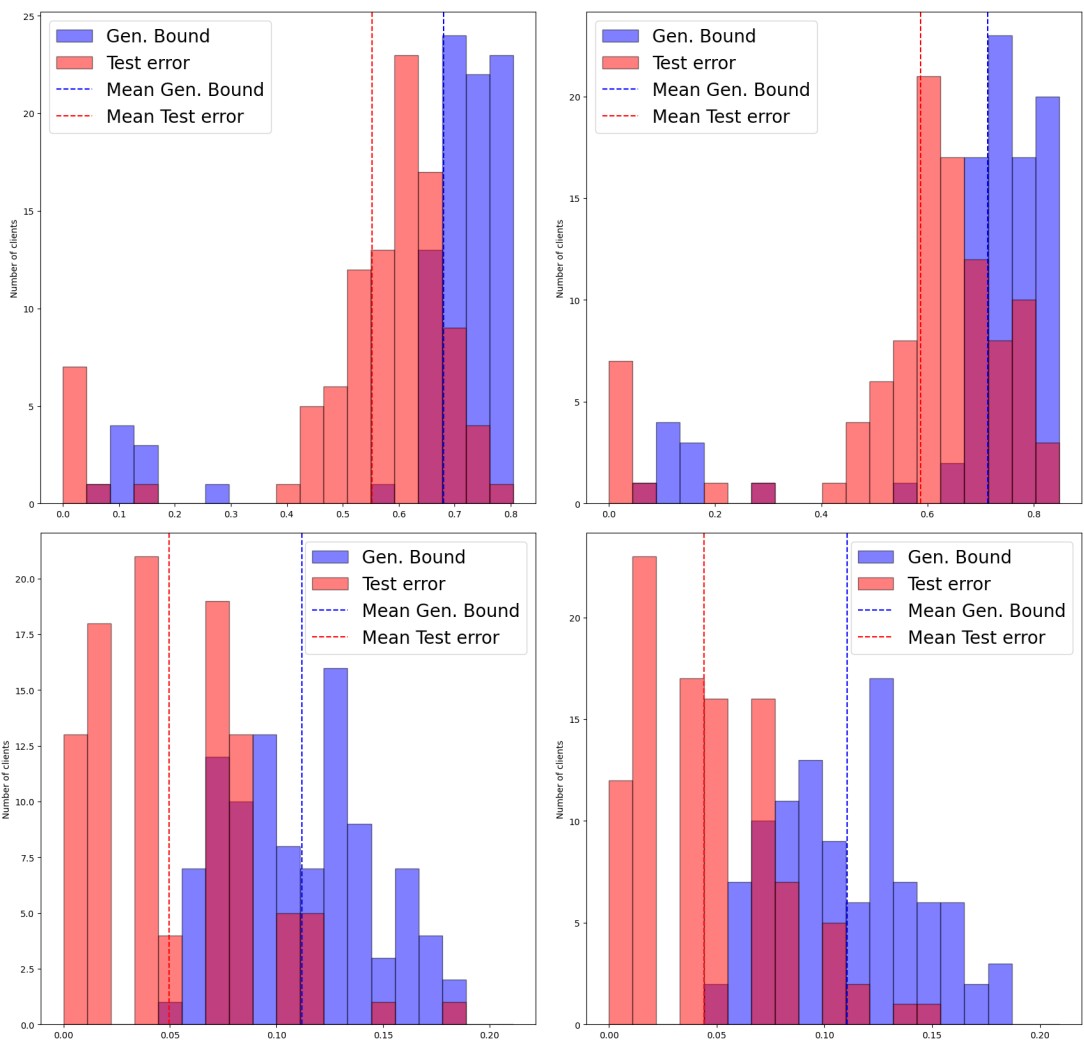

Figure 1: Histograms gathering test errors (red) and bound (blue) of all 100 users of the PFL setting. In order from top to botton: Random prior$-f_1$, Random prior$-f_2$, Learnt prior$-f_1$, Learnt prior$-f_2$.

common prior distribution does not support all users uniformly as we can see in Figure 1. Indeed, our algorithms with random priors suffer from deteriorated bounds and test errors on average and the worst case, but approximately 15% enjoy good test errors and 9% benefit from theoretical guarantees lower than 40%. Also, our algorithms with learnt priors enjoy test errors and generalisation guarantees lower than 20% for all users. Furthermore, approximately half of the users benefit from test errors lower than 5%. This highlights the importance of the prior in this non-*i.i.d.* setting. As the test set of each user is small (60 images), some users achieve a 0% test error.

## 4.2 Classification on CIFAR-10

To evaluate the effectiveness of our algorithm in more challenging scenarios, we conduct experiments on the CIFAR-10 dataset. We followed a setup akin to Section 4.1.1, with minor adjustments. Specifically, we exploit a learned prior trained with 70% of the dataset, the remaining being used for posterior estimation. Given the heightened complexity of the CIFAR-10 dataset, we involve Convolutional Neural Networks (CNNs) with 4 and 9 layers, denoted as CNNet4l and CNNet9l, respectively (these architectures also appear in Pérez-Ortiz et al., 2021, exploited as a baseline). To ensure enhanced performances, we also perform fine-tuning of

hyperparameters related to FL such as local batch size and the number of local epochs, while keeping those associated with PAC-Bayes fixed to ensure relevant comparisons with the baseline.

We conducted a comprehensive grid search across a spectrum of hyperparameters to approximate the optimal model configurations. Specifically, we explored various combinations of local batch sizes $(1, 5, 10, 50)$ and local epoch counts $(1, 5, 10)$. Recognising that 100 rounds of training were inadequate for convergence, we extended the training duration to 300 rounds. Additionally, we adjusted the learning rate by a factor of 10 at round 200 to ease optimisation. From the resulting best configurations, we selected the most promising priors for further investigation. Subsequently, we fine-tuned the learning rate by exploring values between $5 \times 10^{-4}$ and $1 \times 10^{-3}$. Remarkably, we discovered that the optimal learning rate, as reported in Pérez-Ortiz et al. (2021), remains consistent between the centralised and federated settings. Furthermore, we conducted experiments to tune the dropout rate, revealing that a dropout rate of 0.2 yielded the best performance on the CIFAR-10 dataset, consistent with findings in Pérez-Ortiz et al. (2021). This reaffirms the applicability of centralised PAC-Bayes hyperparameters to the decentralised PAC-Bayes paradigm. Detailed results of these experiments, showing accuracies for each experiments on the batch size, epoch counts, learning rate and dropout rate are provided in Appendix D.

Our analysis revealed that the most effective hyperparameter settings were a local batch size of 5 for CNNet9l and 10 for CNNet4l, with a corresponding local epoch count of 1 for both architectures. Notably, increasing the number of epochs led to quicker convergence but marginally reduced accuracy. A learning rate of $5 \times 10^{-3}$ was optimal for both architectures, while a dropout rate of 0.2 was found to be the most effective.

We selected the priors with the highest accuracy from our grid search results (refer to the appendix for accuracy details). Subsequently, the posteriors were trained using identical hyperparameters as their corresponding priors. Additionally, we experimented with the KL penalty technique. Our findings are summarized in Table 3 for comparison with the baseline results presented in the first row. Notably, both the prior and posterior performances are slightly inferior to the baseline. This discrepancy can be attributed to the inherent difficulty of federated learning compared to the batch setting. However, it is crucial to note that the generalisation bounds remain non-vacuous.

Among the configurations tested, the most promising outcome was observed with CNNet9l using a KL penalty of 1.0 in conjunction with $f_2$. This configuration achieved a generalisation bound and a test error of 34.5% and 30.5%, respectively, which is 10% higher than the baseline for both metrics. This a similar conclusion than in Section 4.1, highlighting the price to pay to switch from batch to FL. Surprisingly, the KL penalty trick did not yield an improvement in the generalisation bound, contrary to Section 4.1. This is possibly linked to the intrinsic complexity of CIFAR-10. In particular, the inadequacy of the prior may necessitate further optimisation during posterior training, potentially causing the posterior to diverge significantly from the prior distribution. Consequently, the KL term increases post-training, warranting a more substantial penalty.

## 5 Discussion

In this work, we propose a novel algorithm for FL in two different settings: FL-SOB, which allows to exploit a global generalisation guarantee while keeping data separated; as well as PFL, which only involves an *i.i.d.* assumption for each user's dataset. Our work raises two questions: (a) is it possible to remove the *i.i.d.* assumption?(b) is it possible to maintain a global generalisation guarantee, even in the personalised setting? To answer (a), a line of work first initiated by Kuzborskij & Szepesvári (2019) (for *i.i.d.* data) and continued by Haddouche & Guedj (2023); Chugg et al. (2023); Jang et al. (2023) (for non-*i.i.d.* ones) focuses on PAC-Bayes bounds valid for data distribution with bounded variances. In the PFL setting, this could provide novel generalisation bounds without assuming each user possesses an *i.i.d.* dataset. About (b), the recent work of Sefidgaran et al. (2023) provides elements of answer: they derive a general PAC-Bayesian bound holding for the classical FL setting *for all users simultaneously* involving explicitly the number of users and rounds. This allows fruitful theoretical interpretations (especially on the number of rounds involved during the FL training), but leads to vacuous generalisation guarantees for classification task with a federated SVM. Following another route, based on PAC-Bayes methods for meta-learning, Boroujeni et al. (2024) provides a novel FL algorithm for PFL derived from an original theoretical result with strong performances. However,

| Model | Obj. | $\beta$ | $\epsilon$ | rounds | Bound | Test Error | KL Div | Prior Test Error |
|---|---|---|---|---|---|---|---|---|
| | | | | Pérez-Ortiz et al. (2021) | | | | |
| CNNet9l | $f_1$ | 250 | 100 | "1" | 0.237 | 0.216 | <0.0001 | 0.217 |
| (baseline) | $f_2$ | 250 | 100 | "1" | 0.250 | 0.214 | 0.003 | 0.217 |
| | | | | 100 users - GenFL - KL Penalty=0.1 | | | | |
| CNNet4l | $f_1$ | 10 | 1 | 300 | 0.471 | 0.388 | 0.012 | 0.329 |
| (us) | $f_2$ | 10 | 1 | 300 | 0.469 | 0.391 | 0.011 | 0.329 |
| CNNet9l | $f_1$ | 5 | 1 | 300 | 0.393 | 0.303 | 0.019 | 0.274 |
| (us) | $f_2$ | 5 | 1 | 300 | 0.396 | 0.304 | 0.018 | 0.274 |
| | | | | 100 users - GenFL - KL Penalty=1.0 | | | | |
| CNNet4l | $f_1$ | 10 | 1 | 300 | 0.466 | 0.390 | 0.011 | 0.329 |
| (us) | $f_2$ | 10 | 1 | 300 | 0.458 | 0.391 | 0.009 | 0.329 |
| CNNet9l | $f_1$ | 5 | 1 | 300 | 0.386 | 0.302 | 0.014 | 0.274 |
| (us) | $f_2$ | 5 | 1 | 300 | 0.345 | 0.305 | 0.003 | 0.274 |

Table 3: Table displaying the results for the CIFAR-10 dataset alongside the baseline (batch setting) presented in the first row. 'Prior Test Error' represents the 0-1 error of the prior on the test set. The symbol $\beta$ denotes the batch size of clients, while $\epsilon$ indicates the number of local epochs on each round.

their approach involves distributions on distributions spaces, giving their method a potentially high time complexity, they are also unable to compute nonvacuous generalisation guarantees. This constrasts with our results, even for the personalised setting, at the cost of considering generic PAC-Bayesian bounds, not explicitly tailored for federated learning. Establishing a PAC-Bayes bound designed for FL and leading to a non-vacuous generalisation guarantees remains an open challenge that we aim to address in a future work.

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
