# OpenReview forum: "Federated Learning with Nonvacuous Generalisation Bounds"
_TMLR — Rejected by TMLR_

### Review · Reviewer_rGhX · 2024-06-21

**Summary Of Contributions:**

This paper proposes GenFL (Generalisation-driven Federated Learning), an algorithm in which users try to optimize local PAC-Bayes objectives towards their goal of doing Federated Learning. The authors make their proposal adapted to both the settings of FL-SOB and P-FL and show simple experiments demonstrating the non-vacuity of the output of GenFL.

**Audience:**

Yes

**Claims And Evidence:**

No

**Requested Changes:**

(1)
The paper needs to be entirely rewritten/recalibrated.

Either there needs to be rigorous proof that this algorithm has an analytic generalization bound in terms of the algorithm's parameters - which is then shown to be non-vacuous in some experiments. *Even if one ignores the above -for something to be called a generalization guarantee there at least has to be an analytic expression provable for an upper-bound on the risk of the trained predictor via this method - an upper-bound that makes explicit the dependence of the risk on the distribution output by GenFL and hence the architectural parameters of the net trained."

Or, the paper needs to be formatted as an empirical study about a proposed algorithm - without any claim about a guaranteed non-vacuous generalization bound. But at best a claim that the estimated generalization error of the algorithm's output is non-vacuous - which is what seems achievable by a combination of GenFL and FedBound.

(2)
Equation 6 is quite critical to the whole setup and its self-complete proof needs to be provided here.
This can't just be outsourced to previous work.

(3)
In line 17 of the algorithm how is this function $f_{m,\delta,\mu_{\rm prior},\sigma_{\rm prior}}(w_s^k;b)$  defined? This function has never been defined anywhere else. (I am guessing that it is either of $f_1$ or $f_2$ defined later - but that needs to be much more clearly stated - the shift in notations is very confusing!)

(4)
The dimensions of $\mu_{\rm prior}$ and $\sigma_{\rm prior}$ needs to be mentioned.

(5)
There needs to be some explanation about whether the $m_i$s can be arbitrary or not.

(6)
In the experiments, the way the hyper-parameter search has been done is not that important to know. But it would be much more clear if tables were given of the final chosen values at which experiments are being reported.

(7)
Equation 6 has nothing to do with the setup of FL! Then why is the algorithm FedBound claimed to compute that equation 6? And is this provable or is that also an empirical claim?

(8)
There is a particular way the algorithm FedBound is computing the "final" generalization bound ``Up-bound". This computation needs at least a heuristic proof of correctness. Referring it back to equation 6 is not working for the reasons explained above.

The setup of the paper must define something like "posterior risk of PAC-Bayesian FL" and argue that the output of FedBound is trying to approximate it. Currently, none of this is clear.

(9)
I guess that the FedBound needs to get the output of GenFL sent to it as an input and also the training data used by each user. But there is no call made to GenFL from inside FedBound and that makes it very confusing!

(10)
It's entirely unclear as to why Section 3.2 is so rushed and Algorithms 3 and 4 are in the Appendix. If they constitute the P-FL setup then that needs to be in the main paper and explained as such.

(11)
The first line of Appendix C.2.2 seems to have a missing reference.

(12)
Lastly, it needs to be noted that the submission has a strange format whereby the supplementary file is a copy of the full paper instead of just being the appendices. This definitely needs to be changed in any further revisions.

**Strengths And Weaknesses:**

I am quite new to Federated Learning but largely familiar with the PAC-Bayes literature.  From that limited point of view, the proposal here does look interesting - that one does FL in a way that the measured PAC-Bayes bound at the end of the experiment is non-vacuous. So, limited to that, there is a novel potential in this work. It needs to be noted that the paper makes no attempt to introduce FL to readers like me who have very little familiarity with it. It would be great if a revised paper gives a rigorous mathematical introduction to the idea of FL and how risks are defined there.

 However, the key issue with this paper is that its main claim looks entirely misplaced. At many places, like on the top of page 5 we find sentences like ``We stress that GenFL benefits from nonvacuous generalisation guarantees". But this is not at all what is happening in this paper.

 This algorithm GenFL can be said to be provable only if the risk in the distribution parameterized by $w_T$ has an analytic bound in terms of all the hyperparameters like $T, E, B, m_is,\mu_{\rm prior},\sigma_{\rm prior}$ etc. But such a theorem has never been proven here!

 So one cannot claim that this algorithm has any provably good property or a "generalization guarantee" - let alone any claim about non-vacuity of the bound. So this is at best an empirical study of the measured generalization error of this $w_T$ output being non-vacuous in certain experiments - and not a proposal with guarantees.

---

### Review · Reviewer_7seU · 2024-07-17

**Summary Of Contributions:**

The paper introduces a novel strategy for training randomized predictors in federated learning (FL) that aims to preserve privacy while maintaining predictive performance. The authors propose a global randomized predictor based on PAC-Bayesian generalization bounds that inherits the properties of local private predictors. They address both synchronous and heterogeneous cases, demonstrating through numerical experiments that their approach can achieve performance comparable to batch learning. The proposed method, GenFL, is applicable in both FL-SOB and PFL settings, offering nonvacuous generalization guarantees and highlighting the trade-offs between privacy and performance.


The proposed method iaddresses federated learning by integrating PAC-Bayesian theory to achieve a balance between privacy preservation and predictive performance. The GenFL algorithm is meticulously designed to work in both synchronous and personalized federated learning settings, offering robust theoretical foundations through PAC-Bayesian generalization bounds to ensure nonvacuous guarantees. Despite the innovative contributions and comprehensive experimental validation, there are areas for improvement, such as addressing the limitations of the IID assumption, optimizing communication efficiency, and enhancing experimental validation on larger and more diverse datasets. By addressing these aspects, the method could significantly advance the field of privacy-preserving federated learning, making it more applicable and effective in real-world scenarios.

**Audience:**

Yes

**Broader Impact Concerns:**

N/A.

**Claims And Evidence:**

Yes

**Requested Changes:**

Address IID Assumption: Provide a discussion on how the method can be adapted or extended to handle non-i.i.d. data distributions common in real-world federated learning scenarios.
Expand Generalization Analysis: Include a detailed analysis of how PAC-Bayesian generalization bounds perform under non-i.i.d. conditions and discuss potential limitations.
Enhance Experimental Validation: Conduct experiments using GPUs to handle the heavy mathematical computations more efficiently. Additionally, use larger and more diverse datasets to better demonstrate the method's ability to improve model generalization.

**Strengths And Weaknesses:**

Strength
1. The method effectively balances privacy and predictive performance in federated learning using PAC-Bayesian theory.

2. The method provides a theoretical foundation by leveraging PAC-Bayesian generalization bounds to ensure nonvacuous guarantees.

Weakness
1. The assumption that datasets are independent and identically distributed (IID) may not hold in many real-world federated learning scenarios, potentially limiting the generalizability of the results.

2. The reliance on PAC-Bayesian generalization bounds may not fully capture the complexities and variabilities present in non-i.i.d. data distributions, potentially affecting the robustness of the method.

3. The experiments are conducted using only CPUs, which can be inefficient given the heavy mathematical computations involved. Additionally, the datasets used are relatively small, which may not fully demonstrate the improvements in model generalization.

---

### Review · Reviewer_eetV · 2024-07-18

**Summary Of Contributions:**

This paper introduces a novel strategy to train randomised predictors in federated learning where local predictors can preserve privacy while the global predictor inherits the properties of the local private predictors in the sense of a PAC-Bayesian generalisation bound. Both the synchronous case and the heterogenous and homogenous case are studied. Numerical experiments show that GenFL achieves a comparable predictive performance to that of batch approach.

**Audience:**

Yes

**Broader Impact Concerns:**

No concerns on the ethical implications of the work.

**Claims And Evidence:**

No

**Requested Changes:**

1. Thanks for summarizing a lot of related work on page 2 but the contributions of this paper is not clear. What are the improvements made in this paper compared with existing work? What I can find are just a new algorithm and its comparable performance.
2. On page 2, by saying "we show a global generalisation bound for all users in FL-SOB, and local ones in PFL", where are these bounds? There is no bound in Section 3. If they are the same as ones in Section 2, they cannot be claimed as contributions.
3. On page 11, by saying "our work raises two questions", how these two questions raised in this work had been studied by exiting work? Also, contributions of this work should be highlighted again in Section 5 before discussing potential future directions.

**Strengths And Weaknesses:**

Strengths:
1. This paper studies training randomised predictors in federated learning, which is an important problem in federated learning research area.
2. The GenFL approach is well motivated by generalisation bound which has strong theoretical foundation.
3. Section 2 Background is well written and covers a lot of important work in this area.
4. Code is shared anonymously for reproduction of results.

Weaknesses:
1. Contributions of this paper are not clear to me. Clarification is needed, shown below.

---

### Review · Reviewer_gDH2 · 2024-09-07

**Summary Of Contributions:**

This paper adapted the SOTA PAC-Bayes algorithm to the fedarate learning settings and also managed to compute tight global and local generalization bounds for each user in the system. The authors implemented some experiments to compare with the non-FL algorithm in Pérez-Ortiz et al. (2021) and showed that with KL penalty, their proposed algorithms had almost the same performance as the one in Pérez-Ortiz et al. (2021).

**Audience:**

Yes

**Claims And Evidence:**

Yes

**Requested Changes:**

1.Please explain more about the main novelty of the algorithm GenFL. It seemed that it just added a federated learning aggregation part to the existing PAC-Bayes algorithm with a batch setting.

2.This paper proposed two new PAC-Bayesian learning objectives based on local datasets, but it would be better to explain more precisely when to use f_1 and f_2, respectively.

3.Why the generalization error is nonvacuous when it is smaller than 1? Can the authors provide some evidence or reference?

4.Please write down the KL penalty term explicitly. What does it penalize?

5.What is the global performance in PFL setting? There is also no comparison with other algorithms.

**Strengths And Weaknesses:**

**Strengths:**
This paper provided a comprehensive background introduction and motivated their work well. They effectively adapted the batch PAC-Bayes algorithm to the fedarate learning setting and implemented experiments on benchmark datasets.

**Weaknesses:**
1.Although the proposed GenFL algorithm had comparable performance, I am concerned about the novelty of GenFL.

2.In PFL, the authors only changed m to m_i in Algo 1. I suspect whether the change makes a significant improvement for personalized goals of different nodes. It seems that there will be trade-off between local generalization and global generalization.

3.The comparisons in the experiments seem not sufficient enough.

4.The results in Table 3 seem to suggest a lack of effectiveness of the proposed algorithms.

5.The paper did not show the utility and privacy tradeoff.

---

### Decision · Action_Editor_J3ir · 2024-09-23

**Recommendation:** Reject

**Comment:**

The reviewers requested some reasonable changes but there was no response by the authors. Therefore, the paper can not be accepted in its current form.

**Audience:**

As pointed out by different reviewers, the contribution of the work is not clearly explained, and the experimental evidence is not strong enough for the work here to be of interest to a sub-audience of TMLR.

**Claims And Evidence:**

As pointed out by different reviewers, the paper fails to support several claims and even refers to bounds that are supposed to be introduced but appear nowhere in the paper.